# Do lifestyle factors affect patient reported clinical outcomes after total knee replacement surgery? A feasibility cohort study (PRO-Knee)

Gareth Stephens[1,2]*, Triantafyllos Liloglou[2], Maria Moffatt[3], Chris Littlewood[4]

1 The Royal Orthopaedic Hospital NHS Trust, Birmingham, United Kingdom, 2 Faculty of Health, Social Care and Medicine, Edge Hill University, Ormskirk, United Kingdom, 3 School of Medicine and Dentistry, University of Lancashire, Preston, United Kingdom, 4 School of Health and Society, University of Salford, Salford, United Kingdom

* gareth.stephens@nhs.net

## Abstract

### Aims

To evaluate the feasibility of a substantive cohort study to determine whether modifiable lifestyle factors, including smoking, physical inactivity, alcohol consumption and being overweight, affect patient-reported clinical outcomes after total knee replacement surgery.

### Methods

Adults awaiting total knee replacement surgery were recruited pre-operatively and completed self-reported questionnaires at baseline and 3- and 6-months post-surgery. Feasibility outcomes, including recruitment, retention and response rate of the primary outcome questionnaire were analysed descriptively.

### Results

40 participants were recruited from 183 eligible patients (22%). 87.5% (35/40) participants returned questionnaires at 6-months. 85% (34/40) of participants were overweight (BMI > 24.9), 25% (10/40) drank alcohol (AUDIT-C > 4), 5% (2/40) smoked tobacco and 67.5% (27/40) were physically inactive (GPPAQ classification of 'moderately inactive', or 'inactive').

### Conclusion

Modifiable lifestyle factors including smoking, alcohol use, physical inactivity and being overweight are highly prevalent in patients waiting for total knee replacement. Based on this study, a future substantive cohort study investigating the effect of

**Data availability statement:** The data has been made available in an open data repository, FigShare owned by Edge Hill University, UK: https://figshare.edgehill.ac.uk/categories/Health_sciences/27001. STEPHENS, GARETH; Liloglou, Lakis (2025). The anonymised dataset from: Do Lifestyle Factors affect patient reported clinical outcomes after total knee replacement surgery? A feasibility cohort study (PRO-Knee). Edge Hill University. Dataset. https://doi.org/10.25416/edgehill.30094363.v1.

**Funding:** This study was funded by The Chartered Society of Physiotherapy Charitable Trust (PRF/21/PRE09 to G.S.).

**Competing interests:** The authors have declared that no competing interests exist.

lifestyle factors on clinical outcomes post total knee replacement in the UK NHS is feasible.

## 1. Introduction

Around 110,000 total knee replacement surgeries are undertaken each year within the UK, aiming to relieve the pain caused by osteoarthritis [1]. Whilst many patients are satisfied after total knee replacement surgery, up to 20% of patients are dissatisfied with the outcome of their surgery after one year [2]. This equates to up to 22,000 people in the UK, annually. With the number of total knee replacements expected to rise by 40% by 2060 [3], improving satisfaction and clinical outcomes (pain, function and quality of life) after total knee replacement has been identified as a research priority [4,5]

People who undergo total knee replacement are often in poor health and experience long waiting times for surgery [6,7]. In response to this, many pre-habilitation interventions have been developed and researched to evaluate whether improving the physical health of people prior to total knee replacement, improves outcomes following surgery [8,9]. Despite a plethora of diverse exercise-based pre-habilitation programmes, only short-term improvements in function have been demonstrated post-surgery [10]. Therefore, National Institute of Health and Care Excellence (NICE) (2000) have been unable to recommend that pre-habilitation interventions should be used as standard care in the United Kingdom National Health Service (NHS) [5]. Instead, NICE (2020) recommend that research is required to understand whether interventions targeting aspects of an individual's health, such as lifestyle factors, can improve clinical outcomes after total knee replacement [5]. Given the breadth of potential factors that might influence clinical outcomes following total knee replacement, it is first important to understand which factors should be a priority to address.

Patients awaiting total knee replacement are often provided with information advising them to adopt healthier lifestyles prior to surgery [11]. Both patients and clinical experts agree this is important as it has the potential to reduce complication rates following surgery and improve clinical outcomes [12]. However, it remains unknown to what extent whether lifestyle factors such as smoking tobacco, drinking alcohol, an unhealthy diet (leading to high BMI) and physical inactivity affect clinical outcomes after total knee replacement. If clinicians can identify patients at risk of poor outcomes from total knee replacement due to modifiable lifestyle factors, then targeted pre-operative interventions have the potential to improve outcomes from surgery.

To establish whether lifestyle factors predict outcomes from total knee replacement a substantive cohort study is required. Prior to a substantive, multi-centre cohort study, we aimed to evaluate the feasibility of a future study.

## 2. Materials and methods

The study sponsor was the Royal Orthopaedic Hospital NHS Trust (ROH23ORTH01). A favourable ethical opinion was granted by the North of Scotland Research Ethics

Committee on 13th December 2022 (22/NS/0155) for all study processes including face to face and telephone recorded consent. This paper is reported according to STROBE guidelines for reporting of observational studies [13].

## 2.1 Objectives

The feasibility objectives were as follows:

- The eligibility rate (the number of individuals who met the eligibility criteria as a proportion of those screened).
- The consent rate (the number of participants who consent to be assessed for enrolment, as a proportion of those eligible).
- The rate of recruitment (the number of participants enrolled on to the study as a proportion of those who consent).
- The response rates (number of questionnaires returned at 6-months).

## 2.2 Recruitment and data collection

Recruitment took place between 27th January 2023 and 20 June 2023. Adult patients waiting for total knee replacement were identified by a member of the research team, who screened surgical waiting lists at one elective orthopaedic NHS hospital in the midlands of England for patients who met the eligibility criteria. Patients were eligible for the study if they were adults (>18 years old) waiting for primary total knee replacement due to osteoarthritis. Patients were ineligible if they were waiting for any other knee procedure (including uni-condylar knee replacement or revision surgeries), had been diagnosed with inflammatory arthropathy, had undergone previous alignment surgery on the side of the total knee replacement, or were unable to provide informed consent.

Those, who met the eligibility criteria were sent the study information via post and followed up by phone to discuss the study and confirm the eligibility of patients who wished to participate. Informed consent was taken via post, telephone, or face-to-face appointment depending on individual preference, to ensure inclusive recruitment and optimise participation. Telephone consent was recorded via Dictaphone, with patient permission and the audio file stored on secure password protected drives at the sponsor.

Following consent, patients were enrolled as participants on the study, providing they met one of the enrolment criteria (Table 1), assessed via telephone, or face to face. Patients who provided consent but did not meet the enrolment criteria did not proceed in the study as they did not demonstrate one of the lifestyle factors of interest using validated screening tools [14–18].

Body Mass Index (BMI) is a metric that combines height and weight to produce a value. World Health Organisation categorises BMI into six categories (underweight, normal, pre-obesity/ overweight, obesity class 1, obesity class 2 and obesity

**Table 1. Enrolment criteria.**

| Lifestyle Factor | Screening tool | Enrolment Criteria |
|---|---|---|
| Current smoker | Yes/ No | Smokes tobacco products on 'some days' each week |
| High BMI | Body Mass Index Calculator | Score of > 24.9 |
| Alcohol consumption | Alcohol use disorders identification test for consumption (AUDIT C) | Score of > 4 |
| Physical inactivity | General practice physical activity questionnaire (GPPAQ) | Categorisation of 'moderately inactive' or 'inactive' |

BMI – Body Mass Index.

class 3) [14]. Individuals with a BMI of over 24.9 were enrolled as participants on the study as they are classified as over-weight and more likely to develop long-term health conditions [15].

The Alcohol Use Disorders Identification Consumption Test (AUDIT C) comprises three questions (scored 0–4) which assess alcohol consumption [16]. A score of five or more is suggestive of alcohol consumption which may lead to alcohol related harm. AUDIT C is a subscale of the Alcohol Use Disorders Identification Test (AUDIT) which includes a further seven questions regarding alcohol related behaviours. AUDIT is utilised, when an individual scores five or more on the AUDIT C. AUDIT categorises the risk of an individual experiencing alcohol related harm as either 'low risk', 'increasing risk', 'high risk', or 'possibly dependant' [16,17]). Those who scored five or more on the AUDIT C test were enrolled to participate in this study and went on to complete the AUDIT questionnaire as part of the baseline assessment for the study.

The General Practice Physical Activity Questionnaire (GPPAQ) Is a 7-item questionnaire which identifies whether an individual is meeting the levels of physical activity recommended in NICE guidelines (2003) [18]. Whilst not validated in a total knee replacement population, it is recommended by NICE as a screening tool to measure physical activity [19]. It categorises an individual as either 'inactive', 'moderately inactive', 'moderately active', or 'active'. Those categorised as either 'inactive' or 'moderately inactive', and therefore not meeting the guidelines, are more likely to develop long-term health conditions [20] and were enrolled for participation into the study.

Additional data regarding health status was recorded using the Functional Comorbidity Index which measures the presence or absence of 18 health conditions which are expected to affect physical function. The index provides a final score which is designed to predict functional status, which could be used as part of secondary analysis of a future substantive study [21].

Participants enrolled onto the study completed self-reported questionnaires pre-operatively (within 12-weeks of surgery), and again at 3-months (+/- 4-weeks) and 6-months (+/- 4-weeks) post-surgery (Fig 1). The extent to which participants were aware of their knee, knee pain and disability, health-related quality of life, anxiety and depression were collected at each time point (Table 2).

The Forgotten Joint Score is a patient-reported outcome measure intended to determine a patient's ability to "forget" about their affected joint after surgery or treatment [22]. It consists of 12 questions and is scored on a 0–100 scale, with higher scores indicating higher levels of awareness. The Forgotten Joint Score was our preferred measure as it does not demonstrate the same ceiling effects as other commonly used patient reported outcomes following total knee replacement, such as the Oxford Knee score [22]. A recent systematic review highly recommended the use of the Forgotten Joint Score, determining that its internal consistency was consistently high (Cronbach alpha >0.9) and test-retest reliability was good or excellent (interclass correlation coefficient (ICC) ≥0.8) in all studies [23]. The minimal important change for the Forgotten Joint Score following total knee replacement is 10.8 points [24].

The Oxford Knee Score is a 12-item patient-reported outcome measure specifically designed and developed to assess function and pain after total knee replacement (TKR) surgery [25]. Each item is scored 0–4, with higher scores indicating less severe symptoms and therefore improved outcomes. The Oxford Knee Score has shown high test–retest reliability (r = 0.92) for evaluating pain and function following knee arthroplasty [25]. The minimally important change score for the Oxford Knee Score is 10.5 points [26].

The EQ-5D-5L is a generic measure of health-related quality of life. It provides a single index value for health status that can be used for clinical or health economic evaluation [27]. The EQ-5D-5L consists of questions relating to five health domains (mobility, self-care, usual activities, pain/discomfort and anxiety/depression) and respondents rate their degree of impairment using five response levels (no problems, slight problems, moderate problems, severe problems and extreme problems). The EQ-5D is NICE's preferred measure of health-related quality of life in adults [28]. The EQ-5D-5L index demonstrates excellent reliability (ICC > 0.92) for measuring health-related quality of life across diverse patient populations [29].

The Hospital Anxiety and Depression Scale is commonly used in research studies to assess anxiety and depression [30]. It comprises 14 items (7 items each for anxiety and depression), with a score ranging between 0 and 21 for the anxiety and depression subscales. Scores between 8 and 10 indicate a moderate presence of symptoms, whereas a score

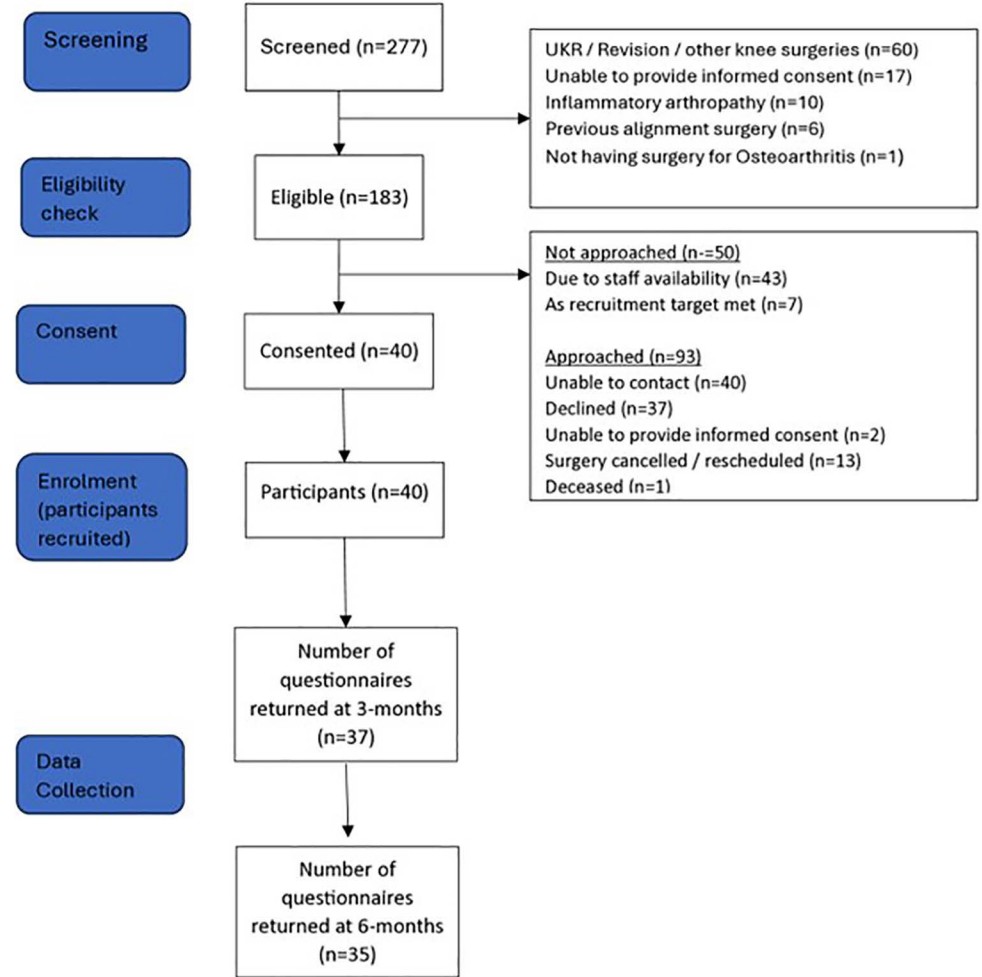

**Fig 1. Flow of participants through the PRO-Knee study.**

greater than 10 indicates a significant number of symptoms that likely correspond with a clinical diagnosis [31]. The HADS has demonstrated good internal consistency (Cronbach's α = 0.87 for anxiety and 0.81 for depression) in detecting anxiety and depression in hospital and primary care settings [30].

### 2.3 Data analysis and sample size

The target sample size was 40 participants which is above the median (n = 30) and on the upper end of the IQR (n = 20–43) of recent (2013–2020) UK pilot and feasibility studies with continuous outcomes, on the International Standard Randomised Controlled Trial Number registry [32]. Feasibility of the study to progress to a substantive cohort study was assessed against pre-determined success criteria with red, amber, green thresholds (Table 3) developed using published guidelines [33]. A summary of how the data from patient reported questionnaires was managed is outlined in supporting information (S1 Table).

Feasibility data are analysed and reported descriptively as feasibility studies are not powered to support inferential statistical comparisons. The exploratory between-group differences in mean change in clinical outcome measures were

**Table 2. Data collected via self-reported questionnaires.**

|  | Baseline | 3-months post TKR | 6-months post TKR |
|---|---|---|---|
| Demographics | • |  |  |
| Work status | • |  |  |
| Forgotten Joint Score | • | • | • |
| Oxford Knee Score | • | • | • |
| EQ5D-5L | • | • | • |
| Hospital Anxiety and Depression Scale (HADS) | • | • | • |
| Alcohol use disorders identification test | • |  |  |
| Functional Comorbidity Index | • |  |  |
| Complications after surgery |  | • | • |
| Satisfaction with:<br> • Pain levels<br> • Walking<br> • Appearance<br> • Return to leisure/ hobbies<br> • Outcome<br>Measured via 5-point Likert scale (very dissatis-fied to very satisfied). |  | • | • |

TKR = Total Knee Replacement.

**Table 3. Success criteria for progression to a substantive cohort study.**

| Success criteria | Red (Stop) | Amber (Amend) | Green (Go) |
|---|---|---|---|
| Consent rate<br>(% of eligible patients who consent to be assessed for enrolment) | <20 | 20 to 29 | 30 or more |
| Recruitment rate<br>(% of consented patients enrolled) | <20 | 20 to 29 | 30 or more |
| Response rates<br>(% of participants returning questionnaires at 6-months) | <70 | 70 to 79 | 80 or more |

undertaken for those with one lifestyle factor, compared to those with three lifestyle factors using Mann Whitney U test ($p < 0.05$).

## 3. Results

### 3.1 Consent and recruitment rates

Of the 183 eligible patients, 40 participants provided consent to be assessed for enrolment on the study, providing a consent rate of 22%, aligning with the amber zone of the success criteria (Table 3). Although 183 participants were sent information regarding the study, only 133 participants were approached to consent due to lack of staff availability (43/183; 23%) and because the recruitment target (n = 40), had been reached (7/183; 4%). Of those who were approached, the consent rate was 30% (40/133), aligning with the green zone from the success criteria. All 40 participants who agreed to be assessed for enrolment, were enrolled as participants on the study, providing a recruitment rate of 100% (40/40), aligning with the green zone on the success criteria. Characteristics of the 40 participants are presented in Table 4.

The most common reasons for non-participation were being unable to contact the participant (n = 40) and those who declined to take part (n = 37).

**Table 4. Baseline characteristics of participants (n = 40 for all characteristics).**

| Sex at birth | Number (percentage) |
|---|---|
| Male | 18 (45) |
| Female | 22 (55) |
| **Age** | |
| Mean Age | 70 |
| Age Range | 49-89 |
| **Employment status** | |
| Employed | 7 (17.5) |
| Retired | 30 (75) |
| Long-term sick/ disabled | 3 (7.5) |
| **Lifestyle factors** | |
| Alcohol only | 2 (5) |
| Alcohol and High BMI | 3 (7.5) |
| Alcohol, High BMI, and Physical inactivity | 5 (12.5) |
| High BMI only | 7 (17.5) |
| High BMI and Physical inactivity | 17 (42.5) |
| High BMI, Physical inactivity, and Smoking | 1 (2.5) |
| High BMI and Smoking | 1 (2.5) |
| Physical Inactivity only | 4 (10) |
| **Co-morbidities (average per participant)** | |
| Co-morbidities | 3.5 |
| **Mental Health (>10 on HADS)** | |
| Anxiety only | 3 (7.5) |
| Depression only | 2 [5] |
| Anxiety and Depression | 5 (12.5) |

HADS – Hospital Anxiety and Depression Scale.

### 3.2 Response rates

The response rate for the study questionnaires was 87.5% (35/40) at 6-months, which aligned with the green zone of the success criteria at 6-months post-surgery (Table 3). The completion rate at baseline was 100% and the response rate at 3-months (37/40; 92.5%) also aligned with the green zone of the success criteria.

### 3.3 Patient reported clinical outcome scores

At 6-months one participant failed to complete both patient-reported clinical outcome measures sufficiently for analysis; leaving a total of 34 completed outcomes for both the Forgotten Joint Score and Oxford Knee Score. Both clinical outcome measures, demonstrated improvements at 6-months from baseline (Table 5).

Participants who returned clinical outcome measures at 6-months reported an average change of 19.2 (SD 21.5) on the Forgotten Joint Score, above the minimal important change of 10.8 [23]. Participants reported an average change of 10.4 (SD 7.8) on the Oxford Knee Score, just under the established minimal important change (10.5) [25].

For those who returned data at 6-months (n = 34) there was no significant difference in the mean change scores of those with three lifestyle factors compared with those with one lifestyle factor (Table 6) on the Forgotten joint score (u = 32.5; z = −0.05; p = 0.96), or the Oxford Knee Score (u = 31; z = −0.2; p = 0.84). Data for all participants is reported in Supporting information (S2 Table).

**Table 5. Mean scores for clinical outcome measures.**

| Clinical outcomes | Forgotten Joint Score | | | |
| | Baseline | 3-months | 6-months | Mean change (Baseline – 6-months) |
|---|---|---|---|---|
| Forgotten Joint Score<br>Number of participants (%)<br>Mean score (SD) | 40 (100)<br>89.6 (8.9) | 37 (92.5)<br>73.3 (25.4) | 34 (85)<br>69.8 (25.6) | 34 (85)<br>19.2 (21.5) |
| Oxford Knee Score<br>Number of participants (%)<br>Mean score (SD) | 40 (100)<br>18.4 (7.5) | 36 (90)<br>27.1 (9.9) | 34 (85)<br>29.5 (10.0) | 34 (85)<br>10.4 (7.8) |

**Table 6. Number of lifestyle factors and clinical outcomes for those returning data at 6-months.**

| No of lifestyle factors reported by participants | Forgotten Joint Score | | | Oxford Knee score | | |
| | Baseline | 6-months | Mean Change (baseline to 6-months) | Baseline | 6-months | Mean Change (baseline to 6-months) |
|---|---|---|---|---|---|---|
| 1 Lifestyle factor<br>N = 11<br>Mean score (SD) | 86.6 (7.5) | 61.1 (26.1) | 25.4 (24.3) | 23.1 (4.5) | 33.9 (7.3) | 10.8 (6.6) |
| 2 Lifestyle factors<br>N = 17<br>Mean score (SD) | 91.3 (8.2) | 78.5 (18.6) | 12.8 (16.5) | 16.7 (6.5) | 26.3 (9.5) | 9.6 (8.0) |
| 3 Lifestyle factors<br>N = 6<br>Mean score (SD) | 87.2(12.2) | 61.0 (36.6) | 26.1 (27.1) | 18.7 (8.0) | 30.5 (13.9) | 11.8 (9.9) |

N = Number of participants.

### 3.4 Satisfaction scores

At 6-months (Supporting information, S3 Table) post total knee replacement, most participants (20/35; 57.1%) were either satisfied or very satisfied with their outcome. Conversely 22.9% (8/35) participants were either dissatisfied or very dissatisfied with the outcomes of their surgery at 6-months.

### 4. Discussion

A substantive cohort study to investigate the effect of lifestyle factors and clinical outcomes following primary total knee replacement is feasible. The pre-defined success criteria were met for recruitment (40/40; 100%) and response rates (35/40; 87.5%) at 6-months.

The rate of consent (43/183; 23%) met the amber zone criteria, suggesting improvements should be considered. However, 50 eligible potential participants were not contacted by the research team following sending of the initial study information due to lack of staff availability (n = 43) and because the recruitment target had been met (n = 7). The consent rate of those who were approached is 30% (40/133), which aligns with the green zone outlined in the success criteria.

The recruitment rate (40/40; 100%) from this feasibility study suggests that the prevalence of lifestyle factors, which are associated with poor long-term health in patients undergoing total knee replacement could be high. It is not unexpected that 67.5% (27/40) of participants in this feasibility cohort study were categorised as physically inactive (via GPPAQ questionnaire) prior to surgery, as the goal of most patients from knee replacement surgery, is to relieve knee pain during physical activity [34]. However, only 10% (4/40) of participants met the enrolment criteria for physical inactivity alone and therefore most participants had more than one lifestyle factor associated with poor long-term health. The most common

presentation was physically inactivity, combined with a BMI > 24.9 (42.5%). High BMI was the most common lifestyle factor identified in this cohort study, prevalent in 85% (34/40) of participants. Furthermore, on average, participants had at least three (average 3.5) long-term health conditions they were managing at the point they consented to surgery. This cohort study therefore supports current evidence that people undergoing total knee replacement are often in poor health and living with long-term health conditions [6,35]. Future interventions which may target improvement of lifestyle factors such as physical activity would need to consider the multi-dimensional influences on reduced physical activity.

This feasibility cohort study revealed that 22.9% (8/35) of participants were either dissatisfied (n = 5) or very dissatisfied (n = 3) with the outcomes of their total knee replacement at 6-months; with a further 20.0% (7/35) being neither satisfied. This is in keeping with the rates of dissatisfaction (20%) reported in contemporary literature [2].. This feasibility study does not aim and is not powered to make conclusions on the success of total knee replacement surgery, but it raises important questions regarding the success of total knee replacement surgeries for a significant proportion of patients. Patient satisfaction following total knee replacement is a complex construct influenced by clinical outcomes, expectations and patient-clinician relationships [36]. Further research is required to standardise the way satisfaction post total knee replacement is recorded and reported.

The prevalence of lifestyle factors (determined by the recruitment rate) and dissatisfaction data from this study support previous findings that suggest research into the effects of lifestyle factors on outcomes from total knee replacement is necessary. If a future substantive cohort study evaluates that lifestyle factors are associated with poorer clinical outcomes, interventions targeting modifiable lifestyle factors could help large numbers of patients waiting for total knee replacement to improve their outcomes from surgery. In this feasibility cohort study however, there was no signal that any lifestyle factor in isolation, or the total number of lifestyle factors, are associated with poorer clinical outcomes from total knee replacement surgery. The study is not powered to make definitive conclusions; therefore, a future substantive cohort study is indicated.

Consideration should be given to the primary outcome measure used in a future substantive cohort study. In this feasibility study, clinical outcomes, measured via the Forgotten Joint score suggested the average improvement experienced by participants (19.2 points), exceeded the minimal important change [24]. Whereas the scores from the Oxford Knee Score were almost equivalent (10.4) to the minimally important change (10.5 points) at 6-months [26]. The Oxford Knee Score has been shown to be a more sensitive at measuring clinical outcomes in the earlier stages of recovery following total knee replacement [37], whereas the Forgotten Joint Score has been shown to be more sensitive when measuring clinical outcomes of patients who demonstrate higher levels of physical function than the Oxford Knee Score [38]. Furthermore, clinical outcomes at 6-months have been demonstrated as a good indicator of long-term outcome from total knee replacement [39]. However, patients continue to report improvements in their pain and disability at 12-months and beyond [40].

A limitation of the study is that all recruitment was conducted in one single elective orthopaedic hospital, which may not accurately reflect recruitment and retention across other NHS Trusts in the United Kingdom due to the diversity of practices and populations across the UK.

The study used BMI as a proxy for an unhealthy diet which could be modified with intervention. It is recognised that whilst high BMI is a predictor of long-term poorer health, it does not necessarily indicate that people have an unhealthy diet or would benefit from weight loss strategies [41,42]. However, food diaries are prone to inaccuracies and in the context of this cohort study and what is feasible within the NHS, it was felt to be a pragmatic solution [43].

The EuroQol EQ5D-5L was used to measure health-related quality of life, however NICE (2019) do not recommend the use of this tool in the UK, until a validated data set for the UK has been established [44].

## 5. Conclusion

All patients who consented to take part in this study reported at least one relevant lifestyle factor including smoking tobacco, drinking alcohol, being physically inactive, or overweight. This confirms that these lifestyle factors are highly

prevalent in patients waiting for total knee replacement surgery. Based on our findings, A future substantive cohort study to investigate the effect of lifestyle factors on clinical outcomes after total knee replacement is feasible. Given the number of total knee replacement surgery's undertaken each year, the current levels of patient dissatisfaction after the procedure, and the potential effect of modifiable lifestyle factors, this further study is now warranted.

## Supporting information

**S1 Table. Management of Patient reported outcomes data.**
(DOCX)

**S2 Table. Number of lifestyle factors and clinical outcomes for all participants.**
(DOCX)

**S3 Table. Satisfaction with aspects of outcome at 6-months post total knee replacement.**
(DOCX)

## Author contributions

**Conceptualization:** Gareth Stephens, Chris Littlewood.

**Data curation:** Gareth Stephens.

**Formal analysis:** Gareth Stephens, Triantafyllos Liloglou, Chris Littlewood.

**Funding acquisition:** Gareth Stephens.

**Investigation:** Gareth Stephens.

**Methodology:** Gareth Stephens, Triantafyllos Liloglou, Maria Moffatt, Chris Littlewood.

**Project administration:** Gareth Stephens.

**Supervision:** Triantafyllos Liloglou, Maria Moffatt, Chris Littlewood.

**Writing – original draft:** Gareth Stephens.

**Writing – review & editing:** Triantafyllos Liloglou, Maria Moffatt, Chris Littlewood.

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
