## [Decision Letter · Decision Letter 0]

24 Mar 2025

PONE-D-25-04937
Do Lifestyle Factors affect patient reported clinical outcomes after total knee replacement surgery? A feasibility cohort study (PRO-Knee)
PLOS ONE

Dear Dr. Stephens,

Thank you for submitting your manuscript to PLOS ONE. After careful consideration, we feel that it has merit but does not fully meet PLOS ONE’s publication criteria as it currently stands. Therefore, we invite you to submit a revised version of the manuscript that addresses the points raised during the review process.

**ACADEMIC EDITOR: Dear Author, please attend to all comments provided by the reviewers and make necessary corrections.**

We look forward to receiving your revised manuscript.

Kind regards,

Zulkarnain Jaafar

Academic Editor

PLOS ONE

Journal Requirements:

2. Please amend either the title on the online submission form (via Edit Submission) or the title in the manuscript so that they are identical.

3. We note you have included a table to which you do not refer in the text of your manuscript. Please ensure that you refer to Table 4, 6, and 7, in your text; if accepted, production will need this reference to link the reader to the Table.

4. Please remove all personal information, ensure that the data shared are in accordance with participant consent, and re-upload a fully anonymized data set.

Reviewers' comments:

Reviewer's Responses to Questions

**Comments to the Author**

1. Is the manuscript technically sound, and do the data support the conclusions?

Reviewer #1: Yes

Reviewer #2: Partly

2. Has the statistical analysis been performed appropriately and rigorously? 

Reviewer #1: N/A

Reviewer #2: I Don't Know

3. Have the authors made all data underlying the findings in their manuscript fully available?

Reviewer #1: Yes

Reviewer #2: No

4. Is the manuscript presented in an intelligible fashion and written in standard English?

Reviewer #1: Yes

Reviewer #2: Yes

5. Review Comments to the Author

Reviewer #1: Thank you very much for allowing me to review your manuscript. I appreciate the effort you have made performing this study and submitting it to PLOS one.

From my prospective, the sample size is low and it can result in different error in final conclusion. So, the biggest sample size is warranted.

Reviewer #2: Comments on a manuscript submitted to PLOS One entitled:

Do Lifestyle Factors affect patient-reported clinical outcomes after total knee

replacement surgery? A feasibility cohort study (PRO-Knee)

(# PONE-D-25-04937)

General Comment

Thank you for the opportunity to review the above-mentioned submission, which evaluates the feasibility of a large-scale study into whether lifestyle factors affect patient-reported clinical outcomes following a total knee replacement (TKA). If and how different modifiable lifestyle factors affect the outcomes following TKA is a very important question, and I appreciate the author’s efforts to tackle this topic. However, as it stands now, the manuscript needs some revisions before it is suitable for publication. Below, I have provided some comments that may assist the authors in enhancing the presentation of their work.

Specific comments

Introduction

p. 3 Line 70: A TKA aims to relieve pain and restore function/ADL ability and, ultimately, patients’ quality of life. I suggest adding that information.

p. 3 Line 81: Please introduce abbreviations before using them.

p. 3; Line 83: Again: Please introduce abbreviations before using them.

p. 3; Lines 91/92: I do not fully agree with this sentence. Although it may not be fully understood, there is evidence to suggest an association between lifestyle factors and inferior clinical outcomes following total joint replacement. I suggest rephrasing that statement to read less strongly.

Methods

p. 5; Line 138: You referred to Table 1 already in Line 136. So, no need to refer to it here again. In fact, I suggest combining the sentences lines 135/136 and 138/139 to avoid redundancies.

pp. 5 & 6: It may be due to the formatting in the submission portal, but please make sure not to have dangling tables in the manuscript.

p. 6; Table 1: Please add a caption for BMI.

p. 6, Lines 158 and 164: Five instead of 5.

p. 6/7: Please provide data on the validity and reliability of the AUDIT C, GPPAQ, and Functional Comorbidity Index for your population.

p. 7; Line 176: What exactly covers “functional status”? Only musculoskeletal physical functioning, general physical functioning, or even cognitive aspects? Was the Functional Comorbidity Index (FCI) collected only for this study, or is routinely collected data available from the patient’s medical record? If it was collected outside the clinical routine specifically for this study, why was it important to have that data? You do not report it other than as a mean of comorbidities in the subject characteristics table.

p. 7; Line 177: When pre-operatively? On the day of admission? Or earlier? Was that time the same for all subjects?

p. 7; Line 178: Figure instead of Fig.

p. 7; Line 181 to p. 8; Line 205: Please provide data on the validity and reliability of the used PROMS (FJS, OKS, EQ5D5L, and HADS).

p. 7; Line 185: Why did you choose to use the OKS if you already knew it demonstrates a larger ceiling effect in your investigated patient population? There are a lot of other knee-specific PROMs available in the English language.

p. 9; Line 220: Please introduce the abbreviation ISRCTN

p. 9; Line 223: “Date are” instead of “Data is,” as data are always plural.

p. 9; Line 225: What was your preset significance level? Which software did you use for the statistical analysis? What about those with two lifestyle factors? E.g., the 7.5% alcohol & high BMI or the 42.5% high BMI and Physical inactivity? Did you not include them in your analysis? Please revise accordingly.

p. 9, Table 3: I suggest specifying the table title more. Again, Tables and Figures must be self-explaining and understandable even when taken from their context. In addition, were the different percentages for the success criteria in the red, amber, and green groups chosen arbitrarily or based on a specific (statistical) rationale?

Results

p. 10; Lines 250/251: Was there a difference in characteristics between those who agreed to participate versus those who declined to participate?

p. 10/11; Table 4: I suggest completely revising this table to meet scientific reporting standards. In addition, what questions did subjects have to answer regarding their sex/gender? “Male,” as you report, refers to biological sex, whereas “Gender“ is a social construct that should be reported as men/women and may be different from sex at birth. Please be specific in your wording. Moreover, please be consistent in how you list multiple lifestyle factors in your table. Decide whether to use “and” or “&.”

Regarding employment status, did your subjects work full-time or half-time, in blue-collar or white-collar jobs? Based on Table 2, you collected this information via questionnaires, so I guess patients were not just asked single-item questions regarding their work status. Regarding co-morbidities, which were those? These factors could influence lifestyle and thus should be considered carefully both in the results and discussion section.

p. 11; Table 5: I suggest substantially reworking this table to meet scientific reporting standards. It does not make sense to read “number of participants” under “outcome.” Moreover, measures of central tendency and dispersion are usually reported as “mean (SD)” or “mean (± SD)” instead of one above the other. Furthermore, you should combine the information from Tables 5 and 6 and maybe even Table 7 into one table.

p.11; 310: I do not see the EQ5D and HADS scores reported anywhere in the manuscript. What is the reason for that? It seems selective not to report those. Please provide information for these PROMS in the text and the tables.

Discussion

General Comment: I recommend revising the entire discussion substantially. Currently, most parts of the discussion only echo the results section. I do not see the results embedded in the current literature nor really being discussed.

Specific comments:

p. 14, Lines 322/323: You state “….suggesting improvements should be considered.” Yet, you do not discuss which ones (other than having more staff available for recruitment) these may be and what your strategies would be to address/implement such “improvements.”

p. 14, Lines 328/329: Be careful not to make overly strong statements regarding aspects you did not measure. For example, your data do not suggest ‘the prevalence of lifestyle factors associated with poor long-term health in patients undergoing TKA is high.’. You did not analyze associations, did not establish prevalence values, nor did you measure your participants’ health in the long term (neither before nor after the operation). And claiming, based on just one study, which is not even a Systematic Review or Meta-Analysis, that 6 months is a good window to extrapolate outcomes to the long term seems invalid. Furthermore, as you state later, your study was not powered sufficiently to make any conclusions other than, based on your selected criteria, a larger study seems feasible.

p. 14, Line 331: Please add “before surgery” following “physically inactive.” You did not measure it after. In addition, your conclusion is a bit weak, as patients could still be physically active even with knee OA. They could engage in much more than just walking (e.g., aquatic exercises, cycling, fitness training, etc.). In addition, given that subjects had an average of 3.5 comorbidities/long-term health conditions, OA might not be the (only) reason for their physical inactivity. Please discuss this appropriately.

p. 14, Line 336: be consistent in your reporting. Your criteria were BMI >24.9

p. 14; Line 3343-348: this is all just a repetition of your results.

p. 15, Lines 354-356: I would rather say “supports previous findings” instead of “suggests,” as this was already known before this study.

p. 15, Lines 373/374: As stated above, I do not think making this assumption based on one single study is valid.

p.16, Lines 377-379: I agree. In addition, there may be differences regarding the socioeconomic and educational status of residents between different regions of the UK, possibly leading to differences in respective patients’ lifestyles and or health status.

References

Please check your entire reference list for consistency in your citation style. Some journal names are fully written out, while the respective journal’s abbreviation accompanies other references. Some journal names are italicized, others not. Not all references include all necessary information (e.g., #9, where the journal name is missing). Moreover, please provide the last access date with weblinks.

Supplement

S1 appendix: Please include information on the number of cases in which the proposed handling of missing data has been needed. Furthermore, here again, health-related quality-of-life data and data for anxiety and depression are mentioned. However, I could not find such data in the results section or the discussion. Please add the EQ5D5L and HADS data to your manuscript and discuss those sufficiently.

S2 appendix: Please re-work the table to meet scientific standards of reporting (e.g., do not use “No” for the number of participants but “N=..” or “n= ..”; report Mean (SD) as 87.5 (SD 7.8) for easier reading; please provide information regarding score interpretation in a table caption to ease data interpretation, etc.)

6. PLOS authors have the option to publish the peer review history of their article (what does this mean?). If published, this will include your full peer review and any attached files.

Reviewer #1: No

Reviewer #2: No

---

## [Author Response · Author response to Decision Letter 1]

27 May 2025

 The files have been named clearly so that it is clear what they contain.

2. Please amend either the title on the online submission form (via Edit Submission) or the title in the manuscript so that they are identical.

The manuscript has been amended

3. We note you have included a table to which you do not refer in the text of your manuscript. Please ensure that you refer to Table 4, 6, and 7, in your text; if accepted, production will need this reference to link the reader to the Table.

The tables have been amended and are referenced in the text

4. Please remove all personal information, ensure that the data shared are in accordance with participant consent, and re-upload a fully anonymized data set.

 No personal data is presented in the manuscript

Captions have been included after the reference list

---

## [Decision Letter · Decision Letter 1]

11 Jul 2025

PONE-D-25-04937R1
Do Lifestyle Factors affect patient reported clinical outcomes after total knee replacement surgery? A feasibility cohort study (PRO-Knee)
PLOS ONE

Dear Dr. Stephens,

Thank you for submitting your manuscript to PLOS ONE. After careful consideration, we feel that it has merit but does not fully meet PLOS ONE’s publication criteria as it currently stands. Therefore, we invite you to submit a revised version of the manuscript that addresses the points raised during the review process.

**ACADEMIC EDITOR: **Dear author, after review, we believe your paper has potential but requires major revisions before it can be considered for publication. Please carefully address the reviewers’ comments and revise your manuscript accordingly. 

We look forward to receiving your revised manuscript.

Kind regards,

Zulkarnain Jaafar

Academic Editor

PLOS ONE

Journal Requirements:

Reviewers' comments:

Reviewer's Responses to Questions

**Comments to the Author**

1. If the authors have adequately addressed your comments raised in a previous round of review and you feel that this manuscript is now acceptable for publication, you may indicate that here to bypass the “Comments to the Author” section, enter your conflict of interest statement in the “Confidential to Editor” section, and submit your "Accept" recommendation.

Reviewer #3: All comments have been addressed

Reviewer #4: (No Response)

2. Is the manuscript technically sound, and do the data support the conclusions?

Reviewer #3: Yes

Reviewer #4: No

3. Has the statistical analysis been performed appropriately and rigorously? 

Reviewer #3: Yes

Reviewer #4: Yes

4. Have the authors made all data underlying the findings in their manuscript fully available?

Reviewer #3: Yes

Reviewer #4: Yes

5. Is the manuscript presented in an intelligible fashion and written in standard English?

Reviewer #3: Yes

Reviewer #4: Yes

6. Review Comments to the Author

Reviewer #3: After reviewing the manuscript and the reviewer’s comments, I believe the author has satisfactorily addressed all feedback and made the necessary revisions.

Reviewer #4: This single-center clinical study examined the feasibility to determine whether modifiable lifestyle factors, such as physical activity and being overweight, affect PROMs after total knee replacement (TKR) performed for osteoarthritis. The ultimate goal of the investigators is to perform a substantive multi-center cohort study. The objective of the study is highly relevant because, for still poorly known reasons, up to 20% of patients are dissatisfied with the outcome of the surgery. The current analysis included 40 participants, who were contacted before surgery and followed for a postop period of 6 months. Of those who were approached, the consent rate was 30%. The recruitment and response rates were high. Unfortunately, the study has an obvious methodological shortcoming and fails to give the means to assess the feasibility to carry out a definitive multi-center study.

Specific comments

1. Line 393. It is confusing that “all patients who consented to take part in this study reported at least one relevant lifestyle factor, including smoking tobacco, drinking alcohol, being physically inactive, or overweight”. This means that, based on this feasibility study, the planned multi-center study will not have any control group (patients with no modifiable lifestyle factor) for the comparison. Without a relevant control group, the multi-center study will be meaningless.

2. Lines 346-348 and 360-363. Patient satisfaction is a complex issue and there are no easy ways to find clinically relevant answers. Therefore, it is unexpected that the current feasibility study was based only on the evaluation of subjective outcome (that is PROMs) and did not include any objective measurements of clinical outcome (such as the measurement of daily physical activity and the measurement of walking speed as an evaluation of functional capacity). The primary outcome measure should be, as proposed, one of the selected PROMs but there must be relevant objective data to characterize the features of dissatisfied patients.

3. Table I. High BMI is defined as score of > 24.9. Without doubt, obesity is one of the leading factors behind sedentary lifestyle, probably affecting the outcome. However, a slight overweight (such as BMI values of 25-28) hardly are a worrisome risk-factor for middle-aged and older TKR patients due to the natural age-related increase of body weight. Certainly, BMIs of obesity classes certainly are of a concern.

4. Lines 234-238 and 299-302. There was no significant difference in PROMS of those with three lifestyle factors compared with those with one lifestyle factor. However, looking the data of Table 6, those with two lifestyle factors had the worst outcome. If true, is it relevant to analyze the outcome according to the number of different lifestyle factors? Looking the data of Table 4, one option could be to compare patients with high BMI (such as obesity BMIs > 30.0) and physical inactivity to those with non-obesity and non-sedentary lifestyle.

5. Lines 358-359. The investigators state that this feasibility study was not powered to make definitive conclusions. Therefore, a future substantive cohort study is indicated. Indeed, the minimum requirement is that the investigators open the planned design of the multi-center cohort study and provide its power analysis based on the data of the current study. How many patients are needed? Is it a plan to compare patients with and without lifestyle factors?

7. PLOS authors have the option to publish the peer review history of their article (what does this mean?). If published, this will include your full peer review and any attached files.

Reviewer #3: **Yes: **Raihana Sharir

Reviewer #4: No

---

## [Author Response · Author response to Decision Letter 2]

3 Sep 2025

1. Line 393. It is confusing that “all patients who consented to take part in this study reported at least one relevant lifestyle factor, including smoking tobacco, drinking alcohol, being physically inactive, or overweight”. This means that, based on this feasibility study, the planned multi-center study will not have any control group (patients with no modifiable lifestyle factor) for the comparison. Without a relevant control group, the multi-center study will be meaningless.

We disagree with this review point and suspect the reviewer has misinterpreted our manuscript and the aims of our feasibility study. Objective 1 clearly states that we wanted to explore ‘The eligibility rate (the number of individuals who met the eligibility criteria as a proportion of those screened).’ It just happens that all those who consented had one or more of the relevant lifestyle factors. This is something we wanted to explore at this feasibility stage and does not mean that a future, definitive study, would not include participants without any of the modifiable lifestyle factors.

2. Lines 346-348 and 360-363. Patient satisfaction is a complex issue and there are no easy ways to find clinically relevant answers. Therefore, it is unexpected that the current feasibility study was based only on the evaluation of subjective outcome (that is PROMs) and did not include any objective measurements of clinical outcome (such as the measurement of daily physical activity and the measurement of walking speed as an evaluation of functional capacity). The primary outcome measure should be, as proposed, one of the selected PROMs but there must be relevant objective data to characterize the features of dissatisfied patients.

While we acknowledge the complexity of patient satisfaction, we wish to clarify that our study did not aim to measure satisfaction directly, but instead focused on clinical outcomes—specifically pain, function, and quality of life—using validated patient-reported outcome measures (PROMs).

PROMs are widely used in orthopaedic research and clinical practice as reliable and meaningful tools for assessing outcomes that matter most to patients. Therefore we do not agree that objective measurements are required to obtain clinical outcomes for patients, whilst also reducing significant cost and participant burden.

3. Table I. High BMI is defined as score of > 24.9. Without doubt, obesity is one of the leading factors behind sedentary lifestyle, probably affecting the outcome. However, a slight overweight (such as BMI values of 25-28) hardly are a worrisome risk-factor for middle-aged and older TKR patients due to the natural age-related increase of body weight. Certainly, BMIs of obesity classes certainly are of a concern.

This review point is speculative, and we confirm that we have adopted the World Health Organization’s classification system, which is the most rigorous and defendable approach we currently have.

4. Lines 234-238 and 299-302. There was no significant difference in PROMS of those with three lifestyle factors compared with those with one lifestyle factor. However, looking the data of Table 6, those with two lifestyle factors had the worst outcome. If true, is it relevant to analyze the outcome according to the number of different lifestyle factors? Looking the data of Table 4, one option could be to compare patients with high BMI (such as obesity BMIs > 30.0) and physical inactivity to those with non-obesity and non-sedentary lifestyle.

Whilst it may initially appear that participants with two lifestyle risk factors had worse outcomes than those with three, we believe this is likely due to the small sample size and resultant variability and so there is a risk of over inference if further ‘cherry-picking’ analysis is undertaken.

5. Lines 358-359. The investigators state that this feasibility study was not powered to make definitive conclusions. Therefore, a future substantive cohort study is indicated. Indeed, the minimum requirement is that the investigators open the planned design of the multi-center cohort study and provide its power analysis based on the data of the current study. How many patients are needed? Is it a plan to compare patients with and without lifestyle factors?

Again, we disagree with the reviewer on this point, it would be unwise to report the required sample size of a future definitive study at this stage due to the further design decisions that need to be made.

Importantly, this feasibility work has yielded essential data to inform the design of a future multi-centre cohort study. It has provided estimates of the prevalence of key lifestyle factors (smoking, physical inactivity, alcohol consumption, and overweight) and the variability in outcome measures (PROMs). These data will be used to conduct a formal sample size calculation for the larger substantive study in due course.

To clarify, the aim of the future study is not simply to compare patients with and without any lifestyle factors, but rather to examine the association between specific modifiable behaviours and clinical outcomes but we can confirm that patients without any of the lifestyle factors would be eligible to participate.

---

## [Decision Letter · Decision Letter 2]

7 Sep 2025

Do Lifestyle Factors affect patient reported clinical outcomes after total knee replacement surgery? A feasibility cohort study (PRO-Knee)

PONE-D-25-04937R2

Dear Dr. Stephens,

We’re pleased to inform you that your manuscript has been judged scientifically suitable for publication and will be formally accepted for publication once it meets all outstanding technical requirements.

Kind regards,

Zulkarnain Jaafar

Academic Editor

PLOS ONE

Additional Editor Comments (optional):

Reviewer #4:

Reviewers' comments:

Reviewer's Responses to Questions

**Comments to the Author**

1. If the authors have adequately addressed your comments raised in a previous round of review and you feel that this manuscript is now acceptable for publication, you may indicate that here to bypass the “Comments to the Author” section, enter your conflict of interest statement in the “Confidential to Editor” section, and submit your "Accept" recommendation.

Reviewer #4: All comments have been addressed

2. Is the manuscript technically sound, and do the data support the conclusions?

Reviewer #4: Yes

3. Has the statistical analysis been performed appropriately and rigorously? 

Reviewer #4: Yes

4. Have the authors made all data underlying the findings in their manuscript fully available?

Reviewer #4: Yes

5. Is the manuscript presented in an intelligible fashion and written in standard English?

Reviewer #4: Yes

6. Review Comments to the Author

Reviewer #4: Thank you for the adequate responses and careful revision of the manuscript. I have no further comments.

7. PLOS authors have the option to publish the peer review history of their article (what does this mean?). If published, this will include your full peer review and any attached files.

Reviewer #4: No

---

## [Editor Report · Acceptance letter]

PONE-D-25-04937R2

PLOS ONE

Dear Dr. Stephens,

I'm pleased to inform you that your manuscript has been deemed suitable for publication in PLOS ONE. Congratulations! Your manuscript is now being handed over to our production team.

Kind regards,

on behalf of

Dr. Zulkarnain Jaafar

Academic Editor

PLOS ONE